# Transcriptional Regulation Technology for Gene Perturbation in Fission Yeast

**DOI:** 10.3390/biom13040716

**Published:** 2023-04-21

**Authors:** Ken Ishikawa, Shigeaki Saitoh

**Affiliations:** Department of Cell Biology, Institute of Life Science, Kurume University, Asahi-machi 67, Fukuoka 830-0011, Japan

**Keywords:** knockdown, transcription, CRISPR-Cas, fission yeast, biotechnology

## Abstract

Isolation and introduction of genetic mutations is the primary approach to characterize gene functions in model yeasts. Although this approach has proven very powerful, it is not applicable to all genes in these organisms. For example, introducing defective mutations into essential genes causes lethality upon loss of function. To circumvent this difficulty, conditional and partial repression of target transcription is possible. While transcriptional regulation techniques, such as promoter replacement and 3′ untranslated region (3′UTR) disruption, are available for yeast systems, CRISPR–Cas-based technologies have provided additional options. This review summarizes these gene perturbation technologies, including recent advances in methods based on CRISPR–Cas systems for *Schizosaccharomyces pombe*. We discuss how biological resources afforded by CRISPRi can promote fission yeast genetics.

## 1. Introduction

Fission yeast, *Schizosaccharomyces pombe* (*S. pombe*), is an excellent model organism to help understand the genetic functions of eukaryotes, as efficient forward genetic screening and genetic modification methods are well established. *S. pombe* is useful for revealing chromosomal functions conserved from yeasts to humans and for analyses with cellular visualization; therefore, fission yeast is a preferred model organism to explore broad aspects of cell biology, including chromatin dynamics, cell division, transposable elements, and centromere and telomere regulation [1,2]. Since the complete genome sequence and a genome-wide gene deletion library are available for *S. pombe* [3,4], reverse genetic approaches and systematic functional analyses are also feasible for non-essential genes. However, functional analyses of essential genes are still challenging because of mutational lethality. Traditionally, essential genes have been characterized using conditionally defective mutations such as temperature- and cold-sensitive mutant alleles. However, isolating these alleles requires considerable labor to conduct genetic screening, and even with such effort, it is not always possible to isolate the desired mutant.

Another approach to conditionally perturb essential genes is to tag target genes with auxin-inducible degron (AID), which conditionally facilitates degradation of the target gene product with auxin and a plant F-box protein, although some proteins are less susceptible to this method [5]. Recently, AID, with improved efficiency and reduced side effects, has also become available for *S. pombe* [6]. The other option for gene perturbation that conditionally inactivates the target protein is inducible removal of the target nuclear protein from the nucleus [7]. This method utilizes rapamycin-induced heterodimerization between FRP and FKBP12 proteins. When a target nuclear protein is artificially fused with an FRB domain by gene tagging, this target–FRP fusion protein can be conditionally tethered to abundant cytoplasmic anchor protein fused with the FKBP12 domain by adding rapamycin, which induces dimerization of the FRP and FKBP12 domains. This forces the target nuclear protein to translocate from the nucleus to the cytoplasm, causing loss of function of the target protein. This method was originally established in budding yeast and was designated as an anchor-away technique [7]. The anchor-away technique has also been implemented in *S. pombe*, although whether this technique is generally effective for the nuclear proteins of fission yeast has not been investigated [8]. In animal models, inducible Cre–loxP recombination is utilized for conditional gene knockout in a tissue- and time-specific manner, which can bypass embryonic lethality caused by genetic disruption [9]. Inducible Cre–loxP is available for *S. pombe* and is applicable for the removal of genetic markers and engineering genomic structure [10,11,12], although to our knowledge it has not been utilized for the functional characterization of essential genes.

On the other hand, transcriptional repression, including promoter replacement, 3′-untranslated region (3′UTR) disruption, and CRISPR interference (CRISPRi), of target genes is an additional option to characterize the functions of essential genes [13,14,15], and these technologies for fission yeast are the subject of this review. Here, we summarize recent advances in transcriptional perturbation technology using CRISPR–Cas in *S. pombe*, and its characteristics are compared with those of conventional methods. Potential applications of genome-wide gene knockdown resources that can be constructed using CRISPRi are discussed. Loss-of-function tools for animal models have been reviewed elsewhere [16].

## 2. Transcriptional Regulation Technologies for *S. pombe*

### 2.1. Promoter Replacement

Traditionally, yeast strains that have episomal or integrated plasmids encoding an inducible target gene have been utilized for conditional knockdown of target genes [17,18]. Replacing the native promoter region of a target gene with a controllable promoter can also conditionally perturb gene transcription [19]. For this purpose, a series of controllable promoter units connected with a drug resistance genetic marker, KanMX6, is available [13]. These employ the *nmt1* promoter, which is inhibited with thiamine, and include the *nmt1* promoter variants P3nmt1, P41nmt1, and P81nmt1, which differ in transcription level upon induction [13,20]. These promoters are applicable for essential genes when their induced transcription level is sufficient for viability. Since promoter replacement requires finding a proper promoter for each target gene and manipulating genomic DNA, its application for high-throughput studies is time consuming.

### 2.2. 3′UTR Disruption

Disrupting the 3′UTR by inserting an antibiotic-resistance marker immediately after the stop codon can decrease mRNA abundance by destabilizing target transcripts. This method was originally established in budding yeast and is designated as DAmP (Decreased Abundance by mRNA Perturbation) [14]. Since the modified target mRNA is constitutively transcribed under control of a natural promoter, knockdown of the target gene is not inducible. Therefore, DAmP mutants can be constructed only when this mutation is not lethal. DAmP has been utilized to construct hypomorphic mutants of essential genes and to conduct systematic genetic interaction analyses in budding and fission yeasts [14,21]. The underlying mechanism of DAmP was investigated in budding yeast [22]. mRNA with 3′UTRs abnormally extended by a mutation at the polyadenylation site was degraded by mRNA surveillance machinery for mRNA degradation with premature nonsense codons. Induced mRNA decay is dependent on the decapping enzyme (DCP1) and 5′-to-3′ exonuclease (XRN1), which are commonly utilized even for normal degradation of mRNA after its deadenylation. Other proteins, UPF1, UPF2, and UPF3, which specifically promote rapid decay of mRNA with premature nonsense codons, are also required for the degradation of mRNA with abnormally extended 3′UTRs. DAmP is compatible with auxin-inducible degradation, and combining them can further enhance reduction of target gene products [23]. Since DAmP requires the modification of genomic DNA to disrupt the 3′UTRs of target genes, its application for high-throughput studies is labor intensive.

### 2.3. RNA Interference

RNA interference (RNAi) is a mechanism of gene silencing that is mediated by double-stranded RNA. RNAi has been utilized to characterize gene functions by perturbation of target genes in animals and plants. *S. pombe* also has RNAi machinery, which is required for formation and maintenance of heterochromatin at centromeres and the mating-type locus [24,25,26]. In addition to such physiological activities, RNAi induced by artificial double-stranded RNAs, which are encoded in plasmids or chromosomes, can also repress target gene expression [27,28,29,30]. These artificial RNAi experiments commonly require RNAi machinery proteins, such as Dicer (Dcr1), RNA-dependent RNA polymerase (Rdp1), and Argonaute (Ago1) [28,29,30]. In some cases of artificial RNAi, heterochromatin formation was employed to silence target genes. Such gene silencing depended on factors required for physiological RNAi-mediated heterochromatin formation, including Chp1, Clr4, and Swi6 [29,30]. Thus, *S. pombe* has functional RNAi machinery, and this can be utilized for artificial gene silencing. However, to our knowledge, artificial RNAi has not been utilized to characterize gene function in *S. pombe* as it has in animal models. This may be because gene silencing with artificial RNAi in *S. pombe* is often inefficient. For instance, artificial RNAi targeting a reporter gene, *ura4^+^*, inserted in several loci in chromosomes showed different degrees of gene expression [29,30]. These differences may have been caused by differences in availability of heterochromatin components that are able to spread toward target reporter genes to support gene silencing when inserted in the vicinity of heterochromatin [30]. Moreover, gene silencing of an *ade6–gfp* fusion gene with harpin *gfp* RNA showed instability—only 19% of the cell population showed clear repression of *ade6* gene expression [30]. This instability may have been caused by gene silencing of the hairpin *gfp* RNA gene [30]. In addition to these disadvantageous features of artificial RNAi in *S. pombe*, these methods require time-consuming construction of dsRNA genes covering large regions of target genes (>~300 bp) [27,28,29,30]; thus, it is not suitable for high-throughput gene knockdown.

### 2.4. CRISPR–Cas Based Gene Knockdown Technology

Clustered Regularly Interspaced Short Palindromic Repeat–CRISPR associated (CRISPR-Cas) is a prokaryotic adaptive immune system that eliminates nucleic acids of invading viruses and plasmids. This system provides RNA-guided nuclease activity derived from an effector module that recognizes and cleaves target nucleic acids. CRISPR–Cas systems are currently grouped into two classes and six types based on genetic constitution [31]. Class 1 systems (types I, III, and IV) produce an RNA-guided nuclease composed of multiple subunits, while class 2 systems (types II, V, and VI) encode a single gene for an RNA-guided nuclease. Since class 2 systems can interfere with target nucleic acids using only two components of the RNA-guided nuclease and a short RNA that is complementary to the target of interest, this class has been broadly utilized for applications in biotechnology. In fission yeast, currently three methods based on class 2 systems have been implemented to perturb target gene expression, as discussed below (Figure 1) [32,33,34,35].

#### 2.4.1. dCas9-Mediated CRISPR Interference

Among CRSIPR–Cas systems, a class 2, type II RNA-guided DNA endonuclease, Cas9, was used to establish the first programmed target DNA cleavage in vitro [36], and it is harnessed for genome editing of many model organisms. To cleave target DNA, natural type II systems require three components: Cas9 DNA endonuclease, crRNA (CRISPR RNA), and tracrRNA (transactivating crRNA) [36]. crRNA contains a targeting sequence of 20–30 nt (known as a spacer sequence) that is complementary to a specific target DNA to be cleaved. tracrRNA is required for the maturation of crRNA and DNA endonuclease activity of Cas9. The Cas9 protein, crRNA, and tracrRNA form a ribonucleoprotein complex that recognizes a specific target DNA and introduces a double-stranded break in the DNA. This specific target recognition is supported by two mechanisms: Watson–Crick base-pairing between the spacer sequence of crRNA and a target DNA and recognition of a 2–5-bp protospacer adjacent motif (PAM) that is situated at the immediate 3′-end of the targeted (protospacer) DNA sequence [36,37].

Artificial chimeric RNA constituted with fused crRNA and tracrRNA, which is designated as small guide RNA (sgRNA), provides robust cleavage activity by forming a complex with Cas9 [36]. Therefore, artificial type II CRISPR–Cas systems require only two components: sgRNA and *cas9* genes. Simple alteration of the targeting sequence of sgRNA changes the DNA cleavage specificity of the Cas9–sgRNA complex, and this has been applied for genome editing in many organisms, including *S. pombe* [38,39,40,41]. In *S. pombe*, to produce sgRNA of a precise length, sgRNA is transcribed from the *rrk1* promoter as a fusion with a 5′-leader sequence that will be cleaved by endogenous RNase activity and with a 3′-hammerhead ribozyme sequence that will be removed by self-cleavage [39]. The resulting sgRNA successfully supports target-specific DNA cleavage for genome editing [39].

In addition to genome editing, Cas9 has been harnessed to manipulate transcription of target genes. Cas9 has two DNA endonuclease domains, the HNH (or MrcA) and RuvC-like domains, and each domain cleaves one strand of the target double-stranded DNA. Mutations of catalytic residues in these domains (D10A for HNH domain and H840A for RuvC-like domain) disrupt DNA endonuclease activity [36,42]. A catalytically inactive mutant of Cas9, designated as dead Cas9 (dCas9), recognizes a specific target DNA sequence as normal Cas9 protein does, but interestingly, dCas9 inhibits transcription when it binds to a gene without DNA cleavage. This technology, called CRISPR interference (CRISPRi), was originally established in bacteria [15,43]. The underlying mechanism of CRISPRi was partially investigated in *E. coli*, showing that collision of dCas9 with RNA polymerase directly blocked target transcription [15]. Since conventional CRISPRi does not require any endogenous machinery to repress target transcription, this method is applicable to a broad range of organisms.

In *S. pombe*, we recently implemented this technology using a plasmid encoding an sgRNA and a *dcas9* gene (Figure 1A) [33]. Modification of the targeting sequence in the sgRNA gene was conducted by traditional restriction cleavage and ligation, in which ligation efficiency to produce the expected construct was more than 70% because the golden gate restriction site was employed. A detailed protocol of plasmid construction and CRISPRi induction has been previously described [33]. Since the dCas9 gene was placed after a controllable promoter, *nmt1-41*, CRISPRi could be induced with desired timing by removing thiamine from the medium. In fact, CRISPRi for *ade6^+^* repressed gene expression of the target caused adenine auxotrophy in a medium without adenine and reddish colonies on an indicator medium. These colonies accumulated a red compound in cells in which *ade6^+^* expression was repressed in an inducible manner. Quantification of *ade6^+^* mRNA showed its reduction by as much as 87%. This inducible gene knockdown of *ade6^+^* was reversible. Reddish colonies, formed after CRISPRi for *ade6^+^*, were isolated and grown in the presence of thiamine to repress dCas9 expression, and this resulted in white colonies that indicated abundant expression of *ade6^+^* [33].

As described above, inducible dCas9-mediated CRISPRi was successfully implemented in *S. pombe*. However, the main challenge in CRISPRi is that the efficiency of transcriptional repression depends on where the dCas9–sgRNA complex binds to a target gene. For efficient transcriptional repression by CRISPRi, sgRNA targeting an optimal target region has to be designed. Previous analyses of proper targeting of CRISPRi using bacteria, human tissue cultures, budding yeast, and fission yeast indicated that, in all of these organisms, sgRNAs that bind in the vicinity of the transcription start site (TSS) are usually more effective than those that bind farther away [15,33,43,44]. On the other hand, the influence of the directionality of sgRNAs may be unique for each organism. In *E. coli*, sgRNAs that bound to the non-template strand repressed target transcription more efficiently than those that bound to the template strand, when they were downstream of the TSS [15]. In a genome-wide analysis, this kind of directional preference of sgRNAs was not observed in human tissue culture [44]. In *S. pombe*, we found that (1) sgRNA binding to the non-template strand at the TSS and (2) sgRNA binding to the template strand 60–120 bp downstream of the TSS both provided efficient transcriptional repression by dCas9-mediated CRISPRi (Figure 2) [33]. These sgRNAs caused robust transcriptional repression of model target genes *ade6^+^*, *ura4^+^*, *his2^+^*, and *his7^+^* in *S. pombe* [33]. These rules dramatically reduced the labor needed to design efficient sgRNAs for dCas9-mediated CRISPRi in *S. pombe*, which facilitated systematic transcriptional repression using this method. In fact, these design rules systematically provided effective sgRNAs for uncharacterized essential genes in *S. pombe* [45]. Although the underlying mechanism that causes such sgRNA preferences is still unknown in *S. pombe*, analysis of repression of thousands of genes in human cell lines and budding yeast suggest that nucleosomes impede dCas9-mediated CRISPRi [46,47]. This is consistent with an in vitro observation that dCas9 on DNA was eliminated by histone proteins upon nucleosome assembly [46]. This influence of nucleosomes may impair dCas9-mediated CRISPRi with sgRNA binding around 10–60 bp from the TSS in *S. pombe* (Figure 2). This is consistent with the nucleosome occupancy of this organism [48]. Differences of preferred sgRNA directionality may occur because of differences in the action of transcriptional machinery at the TSS and downstream, namely the establishment of a transcription initiation complex at the TSS and elongation of mRNA in the downstream.

#### 2.4.2. dCas12a-Mediated CRISPR Interference

Class 2, type V CRISPR-Cas systems have an RNA-guided DNA endonuclease named Cas12a (also called Cpf1). In addition to DNA endonuclease activity that cleaves target DNA, the Cas12a protein has RNase activity that processes precursor crRNA to produce mature crRNA [49,50]. Cas12a recognizes a 19 bp direct repeat of a precursor transcript and releases mature crRNA [50,51]. These characteristics of Cas12a offer advantages in its application to genome editing and transcriptional manipulation by enabling a broader array of promoters for crRNA and compact construction of multiplex crRNA genes [32]. As described above, artificial CRISPR–Cas9 requires a specific transcription unit, since sgRNA has to be precisely processed at its 5′ and 3′ ends. Therefore, only promoters having transcripts with discrete cleavage sites are applicable to sgRNA production. In mammals, sgRNAs are usually expressed by an RNA polymerase III-expressed U6 snRNA promoter that starts from a defined G residue and terminates at a poly T (5~6 T) region for artificial CRISPR–Cas9 systems [52]. In Cas9-mediated methods for *S. pombe*, sgRNA is transcribed from an RNA polymerase II-expressed *rrk1* promoter as a fusion of a 5′ leader sequence and a 3′ hammerhead ribozyme, while the other promoter is not available for sgRNA production [39]. However, in the Cas12a-based system for *S. pombe*, other RNA polymerase II promoters are applicable for crRNA production in artificial CRISPR–Cas12a systems because of their RNA processing activity. Since Cas12a has RNA processing activity, gRNA for this enzyme does not require other processing mechanisms as do the leader sequence and ribozyme used in CRISPR–Cas9 systems. In *S. pombe*, this flexibility of promoter choice allowed utilization of the *fba1* promoter, which transcribes more RNA than the *rrk1* promoter, and this increased the genome editing efficiency of the CRISPR–Cas12a system [32]. Another advantage of artificial Cas12a-based systems is that multiple crRNAs can be produced from a single transcript, which are processed by Cas12a to release mature crRNAs [51]. This multiplex crRNA production successfully supported simultaneous genome editing of up to three loci in *S. pombe* [32].

Zhao and Boeke established the first CRISPRi for *S. pombe* using the CRISPR–Cas12a system (Figure 1B) [32]. Cas12a has a RuvC-like DNA endonuclease domain, in which a mutation (D917A in the case of FnCpf1) abolished its DNA endonuclease activity [49]. The DNA-binding activity of this catalytically dead Cas12a (dCas12a) remains RNA-guided, and it was applied to transcriptional repression in *E. coli* [53]. This dCas12a-mediated CRISPRi was implemented in *S. pombe* and showed robust repression of a target gene, *ade6^+^*, causing adenine auxotrophy and reddish colony formation on indicator medium [32]. Among gRNAs tested for the *ade6^+^* knockdown experiment, the one that bound the template strand at the TSS showed the strongest repression of the target. dCas12a-mediated CRISPRi is suitable for such targets around TSSs that have AT-rich sequences since they have T-rich PAMs (5′-TTTV-3′, V = not T) [32]. To identify the design principle for efficient gRNA, it will be necessary to accumulate the results for dCas12a-mediated CRISPRi from different genes, target positions, and gRNA directionalities. It is interesting that dCas9 and dCas12a have different preferences for gRNA directionality in bacteria. In *E. coli*, the dCas9-mediated method was efficient when the gRNA bound to the template strand at the −35 box in a promoter or to the non-template strand when it targeted downstream of the TSS [15], whereas dCas12a showed better repression with the gRNA was complementary to the template strand than to the non-template strand when it was targeted downstream of the TSS [53]. In dCas9-mediated CRISPRi for *S. pombe*, gRNA that binds to the non-template strand is more efficient than gRNA that is complementary to the template strand at the TSS [33] (Figure 2). It is not known whether such gRNA directionality affects the efficiency of dCas12a-mediated CRISPRi in fission yeast. Although dCas12a-mediated CRISPRi with multiplex crRNA has not been attempted in *S. pombe*, this application may be feasible in this organism. A multiplex method successfully repressed target genes in *E. coli*, and this suggests that RNA processing remains in dCas12a, even though its DNA endonuclease activity is defective [53]. This is consistent with the robust effect of dCas12a-mediated CRISPRi in *S. pombe* that requires cleavage of direct repeats of the crRNA precursor (Figure 1B) [32].

#### 2.4.3. Cas13-Mediated Gene Knockdown

Although type II and V systems target DNA, the type VI CRISPR–Cas system has an RNA-guided RNase, named Cas13 (also called C2c2), as its effector module for targeting invading RNA, and this enables gene perturbation applications that degrade target transcripts [54,55]. Type VI systems are classified into four subgroups, A–D [31]. In *S. pombe*, Jing and colleagues first implemented CRISPR–Cas13a (Type VI-A) for gene knockdown, using a plasmid encoding the Cas13a gene under control of an *nmt1* promoter and a crRNA gene transcribed from the *rrk1* promoter (Figure 1C) [35]. This artificial system was attempted for gene perturbation of *tdh1^+^* and *ade6^+^* genes as model targets. Although this method successfully reduced *tdh1^+^* transcripts by ~70%, it showed only a slight effect on *ade6^+^* (~10%). Such tolerance may have been caused by impeded interaction between Cas13a–crRNA and the target mRNA by its secondary structure and/or binding protein. The design principles of crRNA for efficient gene knockdown with this system have not been established. Recently, Cas13d, which is classified as subtype type VI-D, was implemented as a gene knockdown tool for fission yeast [34]. A useful attribute of this system is that Cas13d targets a broader range of target transcripts than Cas13a. Cas13a has a limitation when targeting sequences in vitro and in *E. coli* cells because its activity requires a protospacer flanking sequence (PFS), which is a 3′H (not G) residue situated immediately 3′ of the targeted protospacer sequence [54]. However, Cas13d does not have this sequence constraint [56,57]. Additionally, the Cas13d protein is smaller than Cas13a [31], making it easier to handle. Chen and colleagues established a Cas13d-based gene knockdown method with simultaneous production of multiple crRNAs for single target transcripts in *S. pombe*, using an array of crRNAs that were flanked by two self-cleaving ribozymes (Figure 1D) [34]. A *cas13d* gene driven by a constitutive promoter and a crRNA array gene under control of the inducible promoter *nmt41* were integrated into fission yeast chromosomes. To construct arrayed crRNA, a ~1 kbp, synthetic, double-stranded DNA was inserted into the integration unit. Simultaneous expression of six arrayed crRNAs targeting the *gfp* gene transcript successfully reduced the mRNA level of the target genes fused with a *gfp* DNA sequence, *noc4-gfp*, *bub1-gfp*, and *ade6-gfp* [34]. Thus, Cas13d could inhibit target gene expression without a specific crRNA for a target gene when it was fused with a *gfp* gene sequence. One concern about this method is the instability of repression. When *ade6–gfp* was subjected to knockdown with a *gfp* crRNA array, only 68% of colonies showed reddish color [34]. This suggested that repression of the *gfp*-tagged target may not have been stably maintained. On the other hand, a crRNA array designed for a specific target also successfully reduced target gene expression. Six crRNAs complementary to the 5′-end of the coding region in a target mRNA were simultaneously expressed and provided robust repression of gene expression with Cas13d-mediated gene knockdown. Notably, Cas13 expresses nonspecific nuclease activity after cleaving an RNA-guided specific target transcript [54]. This activity, called collateral activity, is used to detect nucleic acids with high sensitivity [58]. While the existence and/or influence of a collateral effect of Cas13 in cells is still controversial [59], this effect in fission yeast has not been addressed.

## 3. Comparison of Gene Perturbation Technologies in *S. pombe*

Although comprehensive gene knockout mutants of heterozygous diploid deletions are available in *S. pombe*, such a resource involving haploid deletion mutants for essential genes is not available because of lethality, except for haploid spores produced from the heterozygous diploid [4]. Therefore, systematic functional analyses of essential genes in fission yeast are limited. Here, we propose that dCas9-mediated CRISPRi is a desirable option to solve this problem by constructing comprehensive gene perturbation resources for essential genes in *S. pombe*. To construct this resource, several characteristics of the construction method are required: (1) gene perturbation should be conditional; (2) the resource has to be designed systematically so that it can be applied to high-throughput experiments; and (3) gene perturbation should occur on specific targets. The gene perturbation methods summarized in this review article are compared in these respects (Table 1).

Since perturbation of an essential gene can cause lethality, its gene expression should be conditionally repressed. Among the methods described above, DAmP essentially reduces target expression constitutively; therefore, this method is not applicable when it causes lethality. Because of this limitation, DAmP strains targeting essential genes should be hypomorphic. Systematic design of experimental materials is essential to construct comprehensive resources so as to reduce labor. Analyses of gene function by isolating conditionally defective mutants, such as temperature-sensitive mutants, has been a powerful approach for *S. pombe*. However, isolating such mutants to cover all essential genes is currently not feasible because a systematic method is lacking. On the other hand, AID, anchor-away, Cre–loxP, promoter replacement, DAmP, dCas9-mediated CRISPRi, and Cas13d can systematically target arbitrary genes, as described above. However, these methods need to modify genomic DNA by inserting perturbation tags or effecter expression modules, except for the dCas9-mediated method, which are supported by plasmid transformation. Genomic modification is more time consuming than efficient plasmid modification with short oligo DNA insertion. Thus, currently, dCas9-mediated CRISPRi is the most efficient method in terms of throughput. Specificity of gene perturbation is an important feature that influences the fidelity of interpretation potentially obtained from these analyses. Methods that modify genomic DNA, genetic mutations, AID, anchor-away, Cre–loxP, promoter replacement, and DAmP are highly target-specific, except for cases in which the target gene is closely situated to or overlaps with other genes. Transcriptional repression by CRISPRi is also very specific, and off-target effects (direct repression of non-target genes) are rare. Transcriptomic analyses demonstrated that off-target effects of dCas9, dCas12a, and Cas13 are very minor [15,55,60]. Several reports showed that off-target effects of dCas9-mediated CRISPRi are much lower than those of RNAi [61,62,63]. Moreover, mismatches between the targeting sequence of gRNA and target DNA reduce transcriptional repression efficiency [43], and this suggests that off-target repression in CRISPRi is weaker than at the main target. Additionally, the utilization of CRISPRi in organisms with small genomes has an advantage in precision. Cas9 recognizes specific targets mainly with 14 nucleotides (the 2 nt from the PAM and the 12 nt from the seed region that is situated in the gRNA, recognizing the 3′ side of the PAM). In other words, Cas9 recognizes a specific target, with a frequency in random DNA of one per 4^14^ (=268 × 10^6^) nucleotides [64]. Certainly this accuracy may not be sufficiently specific in human cells because of the human genome size (3.1 × 10^9^ bp), in which a 14-nt sequence could exist 11.6 times in the genome, thus requiring careful design of gRNA. However, since the genome size of *S. pombe* (14 × 10^6^ bp) is much smaller, is it usually highly specific. Collectively, in *S. pombe*, dCas9-mediated CRISPRi is conditional, applicable for high-throughput experiments, and highly specific gene perturbation; thus, it is a promising option to construct a comprehensive gene knockdown resource for targeting essential genes.

## 4. Other CRISPR–Cas-Based Technologies on Transcriptional Manipulation

There are several CRISPR–Cas-based technologies that would facilitate fission yeast genetics that have not yet been implemented in this organism. Fusing dCas9 protein with a transcriptional repressor can enhance effects of CRISPRi [65]. Transcriptional repression domains of transcription factors KRAB and Mxi1 have been fused with dCas9, and these fusion proteins provided robust gene perturbation in human tissue culture and in budding yeast [65]. Contrarily, fusion of transcriptional activator domain VP64, which is a 4-copy repeat of the VP16 activator domain, has been utilized to enhance target transcription in human tissue culture [65]. In *S. pombe*, histone H3 lysine 9 methyltransferase Clr4 was artificially fused with dCas12a, and this fusion protein was applied to CRISPRi. Although a dCas12a–Clr4 fusion protein showed stronger target repression than dCas12a, this effect was caused by structural hindrance of Clr4 protein rather than by its histone methyltransferase activity [32]. We tested several transcriptional repression domains and histone modification domains of fission yeast to determine whether they could enhance transcriptional repression using dCas9, but these attempts were unsuccessful (Ishikawa, unpublished results). More searching for linker types and/or fusion orientations may be required to find effective combinations. Manipulating transcription with fusion proteins of Cas and transcriptional regulators in fission yeast is to be implemented in future studies. Importantly, dCas9–KRAB without expression of gRNA affected some parts of the transcriptome in human tissue culture; therefore, careful design of experiments with proper controls is required for reliable interpretation [63]. Another tool with dramatically reduced limitations of CRISPR–Cas technology is Cas variants with relaxed PAM specificity. The original dCas9 and dCas12a require a specific PAM (5′-NGG and 5′-TTTV, respectively) at targeted genes for CRISPRi. However, PAM is not always available at suitable regions of target genes; therefore, Cas with relaxed PAM recognition provides more options for efficient targeting sequences in CRISPRi. Several Cas9 variants with relaxed PAM recognition have been reported [66,67,68], and these variants can potentially solve this issue. Recently, we implemented one such variant, SpG, for CRISPRi in *S. pombe* [45].

## 5. Applications of CRISPRi Resources

CRISPRi with pooled gRNAs that covers a large number of genes has been established as a genetic screening tool for human tissue culture and budding yeast [44,69]. A pooled library was prepared by transformation of the target organism with pooled vectors encoding a variety of gRNA genes. In such systems for human tissue culture, a K562 cell line expressing dCas9 or dCas9 fused with a regulatory domain was transformed with a lentiviral sgRNA pool covering a large number of genes (~16,000), and growth-based screens with this resource identified essential genes, tumor suppressors, and regulators of differentiation [44]. Similarly, in budding yeast, pooled plasmids encoding dCas9-Mxi1 and gRNAs covering the majority of genes were used to transform a parent strain [69]. This library successfully identified adenine and arginine biosynthesis genes by revealing transformed cells exhibiting auxotrophy to these nutrients. These pooled libraries usually detect genes involved in the phenotype of interest by quantification of fitness or abundance of each knockdown strain under particular experimental conditions using deep sequencing [44,69,70]. Because of this technical characteristic, the application of pooled libraries for screening phenotypes other than with fitness quantification is challenging. To conduct phenotypic screenings, a pooled CRISPRi library requires a technical breakthrough such as those established for screening the molecular phenotypes of transcriptional and translational abundance of a gene of interest, quantified with deep sequencing [71]. Thus, pooled libraries are currently not applicable to screen phenotypes that cannot be evaluated with deep sequencing, and those include cell morphological phenotypes (e.g., cell or organelle shape) and cell population phenotypes (e.g., mating-type switch). On the other hand, arrayed libraries with separately maintained strains for each gene knockdown are potentially applicable to any screenings with previously established phenotyping methods for individual genetic analyses. The construction and application of arrayed libraries are difficult for organisms with large genomes because a great amount of labor is required, but it is practical for microbes with small genomes such as *S. pombe*. Arrayed libraries take advantage of microbial models with compact genomes and accumulated phenotyping methods.

## 6. Conclusions

Here, we summarized transcriptional regulation technology available for *S. pombe*. In addition to conventional methods, recent CRISPR–Cas technology is being implemented for this organism. Further characterizing the underlying mechanisms and specificity of this technology will benefit applications. Currently, fission yeast does not have biological resources applicable to comprehensive functional analyses of essential genes; therefore, construction of such resources would greatly promote the understanding of *S. pombe* and eukaryotes. Transcriptional regulation technology can provide technical options to construct such resources. Since fission yeast is suitable for chromosome biology and morphological analyses, exploring these applications with the essential gene knockdown library would offer unique insight that cannot be obtained with traditional resources. Producing metabolites that are valuable in industry or fermentation is another attractive application of this library. The inhibition of an enzyme that catalyzes a reaction in a metabolic pathway can accumulate its substrate, although this expectation has not been practically demonstrated in most metabolic pathways in fission yeast. Such metabolic manipulations are sometimes intractable because ~18% of such enzymes are essential for viability in *S. pombe* (PombBase, https://www.pombase.org/ (accessed on 11 April 2023)). Conditional knockdown of genes encoding these enzymes using the essential gene knockdown library would be able to assess whether target metabolites accumulate upon gene knockdown, and this would facilitate the establishment of methods to produce valuable small molecules by fermentation.

## Figures and Tables

**Figure 1 biomolecules-13-00716-f001:**
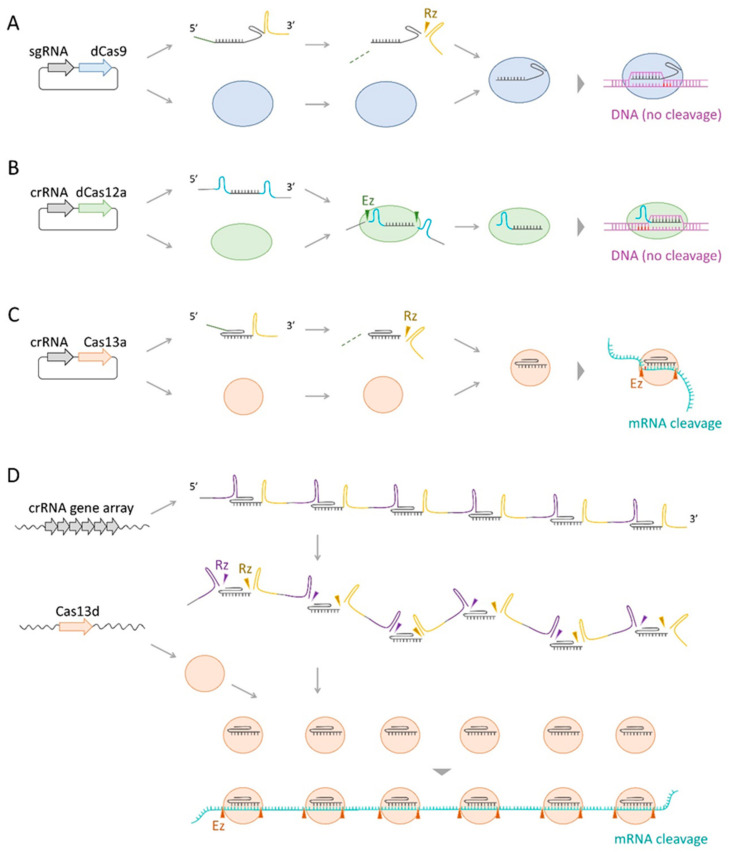
CRISPR–Cas-based gene knockdown technology implemented in *S. pombe.* (**A**) In dCas9-mediated CRISPRi (Ishikawa et al., 2021 [33]), the sgRNA and dCas9 genes are encoded in an episomal plasmid. The sgRNA precursor is processed by removing the 5′-leader with endogenous nuclease activity (dashed line) and a 3′-ribozyme with self-cleavage activity (Rz, ribozyme activity) to produce a mature sgRNA. The sgRNA–dCas9 complex binds to a specific target DNA where it inhibits transcription without DNA cleavage. The dCas9 gene is regulated by an inducible promoter, while the sgRNA gene is driven by a constitutive *rrk1* promoter. (**B**) In dCas12a-mediated CRISPRi (Zhao & Boeke, 2020 [32]), the crRNA and dCas12a genes are encoded in an episomal plasmid. A crRNA precursor is processed by cleaving the 5′-end of direct repeats (light blue) via the nuclease activity of dCas12a (Ez, enzyme activity) to produce a mature crRNA. These cleavages may not be simultaneous. The crRNA–dCas12a complex binds to a specific target DNA where it inhibits transcription without DNA cleavage. Both the cRNA and dCas12a are transcribed by a constitutive RNA polymerase II promoter. (**C**) In Cas13a-mediated gene knockdown (Jing et al., 2018 [35]), the crRNA and Cas13a genes are encoded in an episomal plasmid. The crRNA precursor is processed by removing the 5′-leader with endogenous nuclease activity (dashed line) and a 3′-ribozyme with self-cleavage activity (Rz, ribozyme activity) to produce a mature crRNA. The crRNA–Cas13 complex binds to and cleaves a specific target mRNA (Ez, enzyme activity). The Cas13a gene is controlled by an inducible promoter, while crRNA is driven by a constitutive *rrk1* promoter. (**D**) In Cas13d-mediated gene knockdown (Chen et al., 2023 [34]), the crRNA array and dCas13d genes are encoded in plasmids integrated into *S. pombe* chromosomes. The arrayed crRNA precursor is processed by the self-cleavage activity of two ribozymes adjacent to each crRNA (Rz, ribozyme activity). Each crRNA–Cas13d complex binds and cleaves an mRNA at a specific target region (Ez, enzyme activity). The Cas13d gene is controlled by a constitutive promoter, while the crRNA array gene is driven by an inducible promoter. Length and structure of nucleic acids and proteins presented in this figure do not precisely reflect their molecular properties.

**Figure 2 biomolecules-13-00716-f002:**
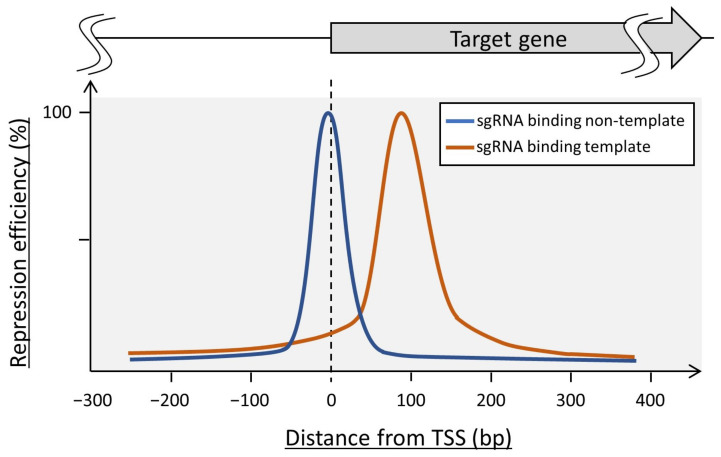
sgRNA design for dCas9-mediated CRISPRi in *S. pombe*. Transcriptional repression efficiency as a function of distance from TSS is shown. Blue and red lines schematically indicate previously reported results of dCas9-mediated CRISPRi for *ade6^+^* and *ura4^+^* genes [33]. Blue line indicates results with sgRNAs that bind to the non-template strand; Red line indicates results with sgRNAs that bind to the template strand.

**Table 1 biomolecules-13-00716-t001:** Comparison of gene perturbation methods implemented in *S. pombe*.

Method	Material Preparation	Conditional Induction	Systematic Design	Relative Throughput	Relative Specificity
genetic mutation	genetic screening	Yes ^a^	No	+	+++
AID	degron tag insertion	Yes	Yes	++	+++
anchor-away	FRB tag insertion	Yes	Yes	++	+++
Cre–loxP	loxP sites insertion	Yes	Yes	++	+++
promoter replacement	promoter replacement	Yes	Yes	++	+++
DAmP	marker gene insertion	No	Yes	++	+++
RNAi	long hairpin RNA (chromosome or plasmid)	Yes	No	++	+
CRISPRi (dCas9)	short oligo DNA cloning (plasmid)	Yes	Yes	+++	++
CRISPRi (dCas12a)	short oligo DNA cloning (plasmid)	No ^b^	No ^c^	++ ^c^	++
Cas13a	short oligo DNA cloning (plasmid)	Yes	No ^c^	++ ^c^	++ ^d^
Cas13d	multiplex crRNA gene cloning (chromosome)	Yes	Yes	++	++ ^d^

^a^, depends on allele; ^b^, potentially applicable for inducible methods by employing an inducible promoter; ^c^, design principle of effective gRNA is not established in *S. pombe*; ^d^, a collateral effect in *S. pombe* has not been evaluated.

## Data Availability

Not applicable.

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
