# Peer review of "Transcriptional Regulation Technology for Gene Perturbation in Fission Yeast"

_biomolecules, 2023, doi:10.3390/biom13040716_

Round 1
Reviewer 1 Report
The authors have done an outstanding job in presenting this topic.
Only one minor spelling correction is required on line 118, of the term “CRSIPR”
The addition of a list of abbreviations is suggested, possibly as an “additional file”, for the benefit of readers who are not familiar with the complex terminology used in this field.
Reviewer 2 Report
The manuscript by Ishikawa and Saitoh reviews gene manipulation techniques that can regulate transcription of genes and help understand function of the gene in fission yeast. The review is quite comprehensive and describes in detail all existing gene manipulation technologies available in fission yeast. They strong point of the review is description of several CRISP-Cas based technologies that have the potential of manipulating genes in fission yeast. They then compare pros and cons of all technologies. The review is an excellent resource for researchers working with fission yeast. The review can be made more effective by incorporating following suggestions.
1. It will be beneficial for a general reader if authors can briefly describe in Introduction all available gene manipulation technologies that are used in different organisms including transgenic mice technology and anchor away approach used in budding yeast. Authors can then discuss the techniques that can be used in fission yeast and the ones that cannot be used and why. These changes will make introduction more effective for a researcher new to fission yeast and to a general reader.
2. Currently, there is only one figure in the review. The review can be made much more effective by including more figures explaining every approach described here. This will especially help readers to understand differences in different CRISPR-Cas approaches described in the review.
3. Minor spelling errors and grammatical mistakes need to be taken care of before submitting manuscript.
Reviewer 3 Report
In this manuscript, the authors summarize the technologies available to reduce the expression of genes in S. pombe and to study the function of essential gene. In particular they focus on CRISPR-based technologies that have been recently developed and compare the efficacy and the applicability of the different strategies to repress transcription in S. pombe.
I found this review comprehensive and well organized. A limited space is dedicated to the "conventional" strategies of transcription repression compared to the CRISPR-based ones, but I understand that it is because of the novelty of the CRISPR-based methods. However, I think that a summary the main advantages and disadvantages of each technique at the end of the paragraph dedicated to the technique can be useful for the readers also to read the table 1. Similarly, I think that a figure depicting the essential elements of the three classes of CRISPR-based systems described in this review will help the readers in understanding the CRISPR-based methods and the differences among them.
Therefore I suggest the following minor modifications of the manuscript:
- Add a summary of advantages and disadvantages for each technique.
- Add a figure reporting a scheme of the three Cas enzyme described and their gRNAs or of the plasmids used for the different CRISP-based methods.
